# Tomography of the source zone of the great 2011 Tohoku earthquake

Yuanyuan Hua [1,2] ✉, Dapeng Zhao [1] ✉, Genti Toyokuni[1] & Yixian Xu [3]

The mechanism and rupture process of the giant 2011 Tohoku-oki earthquake (Mw 9.0) are still poorly understood due to lack of permanent near-field observations. Using seismic arrival times recorded by dense seismograph networks on land and at ocean floor, we determine a detailed seismic tomography model of the megathrust zone beneath the Tohoku forearc. Our results show that the coseismic slip of the 2011 Tohoku-oki earthquake initiated at a boundary between a down-dip high-velocity anomaly and an up-dip low-velocity anomaly. The slow anomaly at shallow depths near the Japan trench may reflect low-rigidity materials that are close to the free surface, resulting in large slip and weak high-frequency radiation. Our new tomographic model can account for not only large slip near the trench but also weak high-frequency radiation from the shallow rupture areas.

[1] Department of Geophysics, Graduate School of Science, Tohoku University, Sendai 980-8578, Japan. [2] Subsurface Multi-scale Imaging Key Laboratory of Hubei Province, Institute of Geophysics and Geomatics, China University of Geosciences, Wuhan 430074, China. [3] School of Earth Sciences, Zhejiang University, Hangzhou, Zhejiang 310027, China. ✉email: huayy2016@gmail.com; zhao@tohoku.ac.jp

The great 2011 Tohoku-oki earthquake (Mw 9.0), one of the greatest earthquakes recorded by modern instruments, occurred in a megathrust zone formed by the subducting Pacific plate and the overriding Okhotsk plate (Fig. 1). It produced a huge tsunami and caused great damage to the local society and infrastructure in NE Japan. The occurrence of this great megathrust earthquake has updated our knowledge about megathrust ruptures, especially due to the large coseismic slip at shallow depth near the Japan trench[1–7] and high-frequency radiations from small-slip areas[8,9]. The mechanism of the large slip near the trench has been interpreted by different models, such as abundance of weak clay (smectite)[10–12] and fault weakening[13]. A depth-variation frictional model was used to account for its energy radiation distribution[14–16]. However, the mechanism and rupture process of the great 2011 Tohoku-oki earthquake are still a matter of some controversy due to the lack of permanent near-field observations[17].

The mechanism of great megathrust earthquakes is influenced by the frictional properties, structure and composition of the plate boundary fault[10]. Hence, mapping the structural variations along the plate boundary fault is critical for understanding the rupture process of this great megathrust earthquake. Results of the Japan Trench Fast Drilling Project (JFAST) suggest that the huge shallow slip may be caused by weak clay (smectite) on the upper boundary of the subducting Pacific plate[10–12,18–20]. However, using residual topography and gravity anomalies, Bassett et al. [21] suggested that the structure and frictional properties of the overriding Okhotsk plate control the interplate coupling and seismogenic behavior in NE Japan. Nishikawa et al. [22] suggest that different slip behaviors exist in the Tohoku forearc, which may be caused by along-strike variations in pore fluids and

lithology of the overriding and subducting plates. Previous tomographic studies[23–25] of the Tohoku megathrust zone suggested that the 2011 Tohoku-oki earthquake hypocenter was located in a significant high-velocity (high-V) anomaly, which may represent an asperity in the megathrust zone. These studies have improved our understanding of the structural variation along the plate boundary. However, all the previous tomographic models have limited spatial resolution in the large-slip area near the Japan trench due to the lack of near-field seafloor observation.

Recently, the Seafloor Observation Network for Earthquakes and Tsunamis (S-net) (Fig. 1) has been installed around the Japan trench and Kuril trench by the National Research Institute for Earth Science and Disaster Prevention (NIED), which consists of 150 ocean bottom seismometer (OBS) stations[26]. The deployment of the S-net provides us with an unprecedented opportunity to image the structure of the Tohoku megathrust zone and to clarify the mechanism of megathrust earthquakes. Using a large number of high-quality arrival-time measurements of local earthquakes recorded by the S-net[27] and Hi-net[28], here we investigate the detailed three-dimensional (3-D) seismic velocity structure beneath the Tohoku forearc, which sheds new light on the mechanism and rupture process of the great 2011 Tohoku-oki earthquake.

## Results and discussion

**Seismic images beneath the Tohoku forearc.** Here we present a new 3-D P-wave tomography model (Fig. 2 and Supplementary Fig. 1) of the Tohoku forearc that is optimized to resolve short-wavelength structures associated with the generation of the great 2011 Tohoku-oki earthquake (for details, see the "Methods"

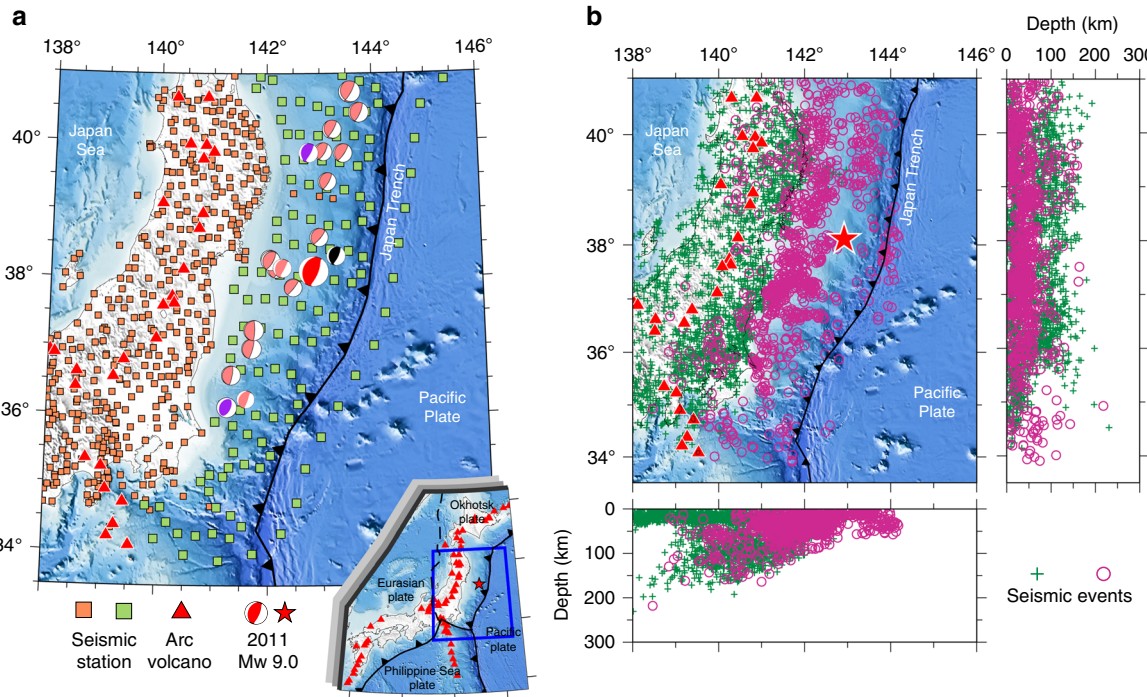

**Fig. 1 Seismic stations and earthquakes used in this study. a** Distribution of the 600 seismic stations used in this study. The orange squares denote 480 seismic stations that belong to the Hi-net. The green squares denote 120 S-net seafloor seismic stations. The red triangles denote active volcanoes. The red beach ball denotes focal mechanism of the great 2011 Tohoku-oki earthquake (Mw 9.0), whereas the pink beach balls denote focal mechanisms of other megathrust earthquakes ($M \geq 7.0$) that occurred during 1917–2017. The black beach ball denotes focal mechanism of the Tohoku-oki earthquake foreshock ($M$ 7.3) on March 9, 2011. The two purple beach balls denote focal mechanisms of two aftershocks ($M \geq 7.5$) of the Tohoku-oki earthquake on March 11, 2011. **b** Distribution of 3757 local earthquakes used in this study. The green crosses denote 2446 events recorded by the Hi-net. The purple circles denote 1311 events that were recorded by the S-net. The red star denotes the mainshock epicenter of the 2011 Tohoku-oki earthquake (Mw 9.0). The jagged black line denotes the Japan trench.

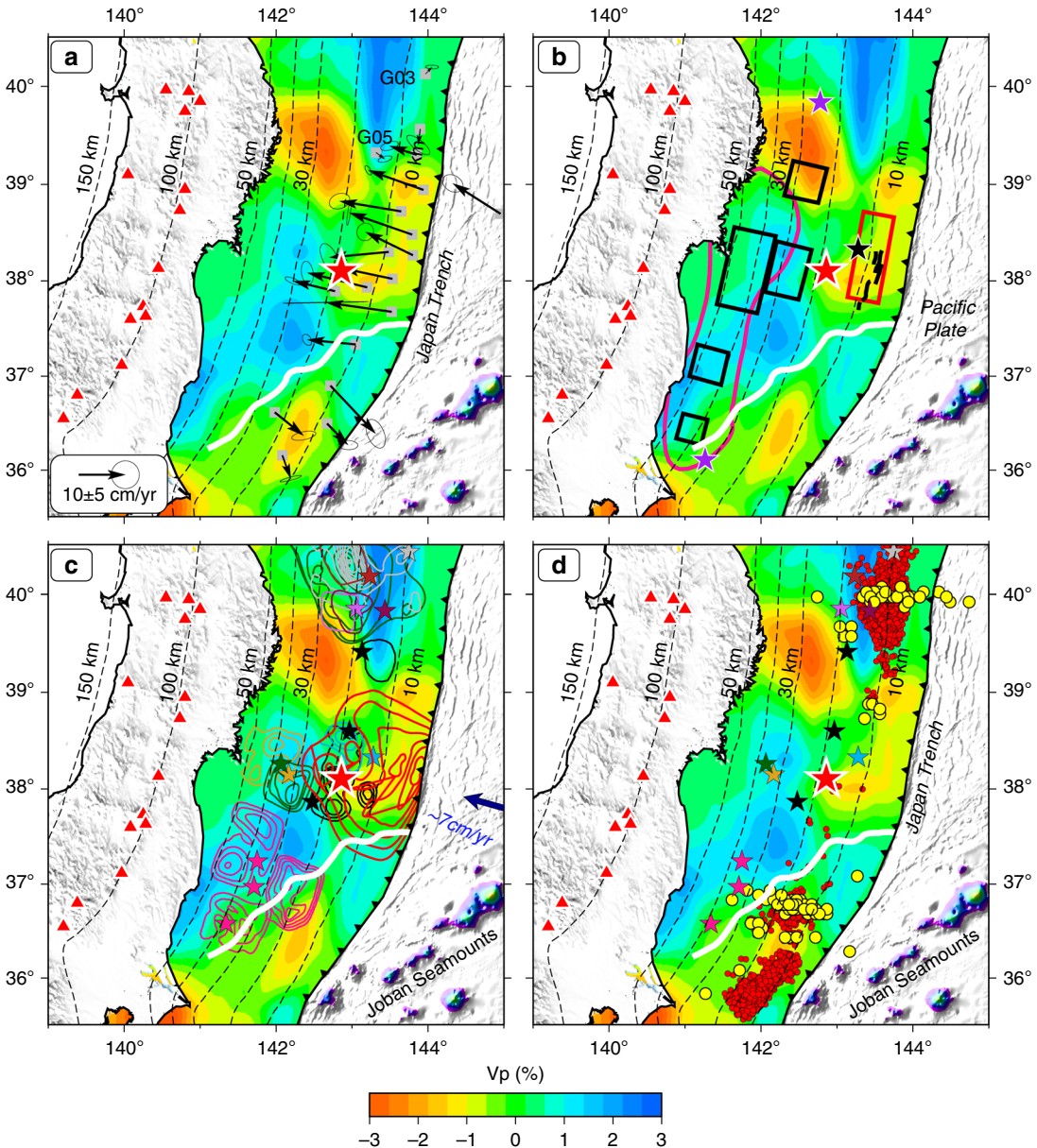

**Fig. 2 Vp tomography and characteristics of the great 2011 Tohoku-oki earthquake.** The colors in **a–d** show residual Vp tomography along the upper boundary of the subducting Pacific plate (black dashed lines) obtained by this study, and the big red star denotes the mainshock epicenter of the great 2011 Tohoku-oki earthquake (Mw 9.0). **a** The black arrows show postseismic displacement rates with 1σ error ellipses estimated by Tomita et al. [44] whose observation periods were from September 2012 to May 2016. **b** The five black rectangles denote locations of coseismic strong ground motions[34]. The magenta contour line marks the site of coseismic high-frequency P-wave radiation with a relatively low seismic moment during the 2011 Tohoku-oki earthquake[37]. The red rectangle denotes a slow slip event (M 7.0) preceding the 2011 Tohoku-oki earthquake[36]. The black short lines denote normal faults near the Japan trench. **c** The red contour lines near the trench denote the coseismic slip distribution of the 2011 Tohoku-oki earthquake[53]. Other color lines and small stars show coseismic slip distributions and epicenters of large megathrust earthquakes[54] ($M \geq 7.0$) that occurred during January 1900 to February 2011. **d** The red and yellow dots denote epicenters of tremors and very low-frequency earthquakes (VLFEs), respectively[22]. The color stars in **d** are the same as those in **c**.

section and Supplementary Figs. 2 and 3). The P-wave velocity (Vp) image along the upper boundary of the subducting Pacific slab (UBP) mainly characterizes Vp variations along the megathrust zone (Fig. 2 and Supplementary Fig. 1), though the Vp image is affected, to some degree, by structural heterogeneities in the upper plate. A two-dimensional (2-D) Vp model above the UBP is shown in Supplementary Figs. 3 and 4, which mainly characterizes average Vp variations in the overriding Okhotsk plate. The most striking feature of our model is along-trench and along-dip Vp variations around the Tohoku megathrust. As

shown by resolution tests (Supplementary Figs. 5–10), our tomographic model is quite robust, even at shallow depth near the Japan trench. We have also conducted a joint inversion of our S-net data and the arrival-time data used by Liu and Zhao [25], and the obtained results (Supplementary Fig. 11) show a pattern very similar to that of Fig. 2 and Supplementary Fig. 1, indicating that our model is stable and reliable.

At depths >20 km, our Vp model (Fig. 2) is similar to the previous tomographic results[23–25], as well as residual topography and gravity distribution (Supplementary Fig. 1). The forearc

segment boundary (FSB) proposed by Bassett et al. [21] is indicated in Fig. 2. From the FSB to ~39°N, our Vp model shows a high-velocity (high-V) anomaly, which corresponds to the high residual topography and positive gravity anomaly. To the north of 39°N, the Vp model, residual topography and gravity all exhibit a negative anomaly where relatively low levels of moderate-size ($M_{JMA} > 5.0$) earthquakes and historical great earthquakes occurred, which was designated as the Sanriku-oki low-seismicity region[29]. At shallow depths <20 km, our Vp model (Fig. 2 and Supplementary Fig. 1) along the megathrust shows a low-velocity (low-V) anomaly from the 2011 mainshock (Mw 9.0) epicenter to the Japan trench, which corresponds to the region of large coseismic slip and lack of high-frequency radiation there. The mainshock hypocenter is located at the boundary between the down-dip high-V anomaly and the up-dip low-V anomaly (Fig. 2 and Supplementary Fig. 1).

The S-net stations were deployed after the 2011 Tohoku-oki earthquake (Fig. 1), hence the tomographic images of the forearc area (Fig. 2) obtained by this study reflect structures after this great earthquake. The frictional heat released during the 2011 coseismic slips could affect seismic velocity along the megathrust zone. However, Fulton et al. [11] showed that the amount of frictional heat generated by the Tohoku-oki rupture was extremely small. In addition, seismic velocity of rocks and minerals is mainly controlled by the lithology of materials at the shallow depth[30]. Hence, we think that the rupture of the 2011 Tohoku-oki earthquake could not cause a significant effect on the velocity structure of the megathrust zone. The small changes of seismic velocity structure caused by the 2011 Tohoku-oki rupture could not be imaged due to the limited spatial resolution of our tomographic model. Therefore, we deem that the main structural features revealed by this study are the same as those before the 2011 Tohoku-oki earthquake.

**Low-velocity anomaly in the large-slip area.** One of the most striking characteristics of the great 2011 Tohoku-oki earthquake is its large coseismic slip at the shallow depth near the Japan trench[5,6,31]. The large slip near the trench has been attributed to dynamic weakening processes, such as thermal pressurization[13,32] and powder lubrication of newly formed gouge[33]. The JFAST drilled into the Tohoku megathrust in the large coseismic slip area, which revealed clay minerals with low friction and low shear stress in the shallow portion of the megathrust zone[10–12]. This suggests that a shallow asperity or a strong patch may not exist near the trench[25]. The presence of low-friction clay minerals (e.g., smectite) may play a key role in the dynamic weakening processes. Recent numerical simulations[32] suggest that the low shear strength of weak clay minerals is insufficient to arrest the inertial motion of rupture propagating along the megathrust, and so large slip could occur near the trench axis.

Our tomographic model (Fig. 2) shows that the 2011 mainshock hypocenter is located at the boundary between the down-dip high-V anomaly and the up-dip low-V anomaly. The up-dip low-V anomaly corresponds to the largest coseismic slip area of the Mw 9.0 mainshock (Fig. 2). This correlation appears to be robust, as shown by different coseismic slip models of the 2011 Tohoku-oki earthquake (Supplementary Fig. 12). However, coseismic slip areas of large megathrust earthquakes ($M \geq 7.0$) that occurred before 2011 are mostly located in the deeper high-V and high-Q (i.e., low seismic attenuation) zones with strong ground motions and positive residual gravity[34,35] (Fig. 2c), which are interpreted as strong asperities[23,25] or granite batholiths[21]. For those megathrust earthquakes ($M \geq 7.0$), the up-dip low-V anomaly may have been able to arrest rupture propagation to the shallow part due to their relatively low slip velocities. However,

for the great 2011 Tohoku-oki earthquake, the low-V anomaly at the shallow depth may not have arrested up-dip rupture due to the large strain rate changes resulting from the extraordinary large slip near the hypocenter. The low-V anomaly with low rigidity may cause not only large coseismic slips near the trench but also a deformed upheaval structure at the shallow depth[5]. The low-V anomaly near the Japan trench may be also responsible for a preceding slow slip event (M 7.0) in the source zone[36] (Fig. 2b).

**Low-frequency radiation near the trench.** Another characteristic of the great 2011 Tohoku-oki earthquake is the lack of high-frequency radiation in the large-slip area at shallow depth (Fig. 2b). The sites of strong high-frequency radiation and strong ground motions are mainly located in the deep portion of the Tohoku megathrust zone beneath the Pacific coast where relatively modest coseismic slip occurred[34,37]. The depth-varying source spectrum suggests differences in slip characteristic along the Tohoku megathrust, such as rapid variations in moment release in the down-dip area and smoother sliding in the up-dip area[38]. Depth-dependent frictional models have been used to account for the occurrence of high-frequency energy radiation down-dip and reduced high-frequency radiation in the large-slip area near the Japan trench[14–16]. Our results show that the sources of strong ground motions and strong high-frequency radiation are mainly located in the high-V anomaly, similar to the previous tomographic results[23–25] (Fig. 2b). This high-V anomaly extends from the FSB to 39°N at depths of 20–50 km. The low-V anomaly at shallow depth indicates low-rigidity materials, such as subducted sediments and increase of dynamic pore-fluid pressure[39], which may account for the weak high-frequency radiation during slip of the shallow part of the megathrust fault.

Previous tomographic studies could only resolve the along-trench velocity variations at depths >20 km[23–25]. Due to the use of high-quality data recorded by the S-net, our tomographic model has a much higher resolution than previous models for the near-trench area. Hence, our model reveals the along-dip velocity variations more completely, allowing us to account for the along-dip distribution of energy radiation and changes in the fault property, such as variations in lithology and/or pore fluid content[14–16]. Typical crystalline rocks comprising the volcanic arc are characterized by a relatively high static coefficient of friction and a high Vp, whereas increasing clay content can reduce the coefficient of friction and seismic velocity[21,40]. The along-dip Vp variation is consistent with the depth-dependent frictional models[14–16], and the high-to-low Vp transition may reflect not only the fault friction transition along the megathrust zone but also the lithologic transition in the upper plate.

The great 2011 Tohoku-oki earthquake caused large crustal deformation in not only the coseismic period[7,41,42] but also the postseismic period[7,43–45]. Recent results of sea floor Global Positioning System and acoustic ranging (GPS-A) show distinct land-ward displacement rates in the large coseismic slip area and remarkable trench-ward displacement rates in the south of the FSB (Fig. 2a)[44]. The viscoelastic relaxation and re-locking of the fault are essential processes in generating the landward movement in the large-slip area, whereas afterslip is required to explain the trench-ward movement in the south of the FSB[21,44]. In the large-slip area, most of the GPS-A stations show landward movement and they are located in the low-V anomaly, except for two stations (G05, G03) that are located in a high-V anomaly and show small trench-ward movements (Fig. 2a). This difference may be caused by not only the variations of friction and material properties along the Tohoku megathrust zone[46] but also the lithologic transition in the upper plate.

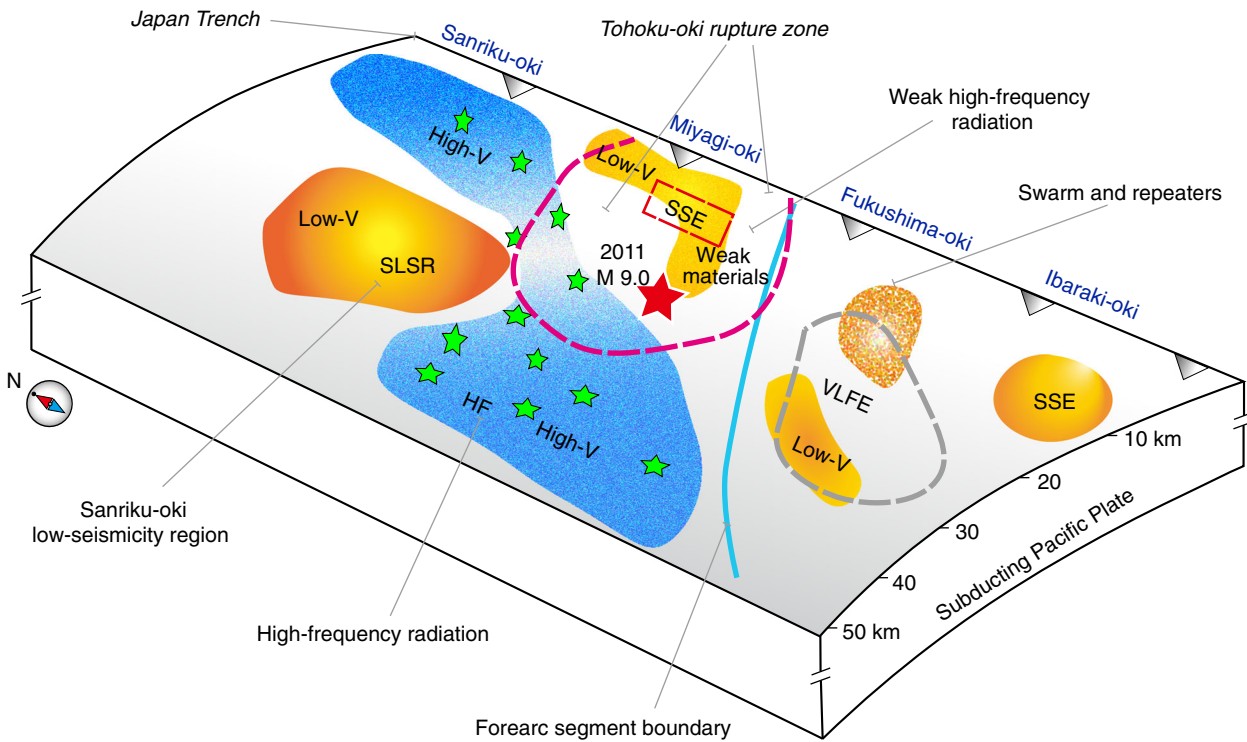

**Fig. 3 A schematic diagram of structural heterogeneities in the Tohoku megathrust zone.** The red star denotes the mainshock epicenter of the great 2011 Tohoku-oki earthquake (Mw 9.0). The green stars represent epicenters of other megathrust earthquakes (M ≥ 7.0). The red dashed line denotes the range of main shock rupture zone of the 2011 Tohoku-oki earthquake. The gray dashed line denotes the area where very low frequency earthquakes (VLLF) occurred. The blue line denotes the forearc segment boundary[21]. SSE, slow slip events. The slab geometry and shape of the Japan trench are simplified.

In summary, our high-resolution tomography sheds new light on the seismogenesis and seismotectonics of the Tohoku forearc, especially the source zone structure of the great 2011 Tohoku-oki earthquake (Fig. 3). Regions of large slip and weak high-frequency radiations near the Japan trench can be directly associated with the low-V anomaly (i.e., weak materials) at the shallow depth. The rupture started at the boundary between the down-dip high-V anomaly and the up-dip low-V anomaly (Fig. 3). The weak materials at the shallow depth could not arrest the strong inertial motion of rupture propagating from down-dip along the megathrust, and so generated the large slips near the trench. The strong high-frequency radiation and strong ground motions mainly originated in the high-V anomaly in the deeper portion of the Tohoku megathrust zone. The large-slip area generally shows weak high-frequency radiation plausibly due to low strength of the shallow low-V anomaly. Hence, near-field seafloor observations such as the S-net enable us to clarify the seismogenesis and seismotectonics in the forearc region and generation mechanism of great megathrust earthquakes, providing crucial information for seismic hazard assessment.

## Methods

**Seismic tomography.** In this work, we used high-quality P-wave arrival-time data of local earthquakes from June 2002 to December 2018 recorded at 480 permanent stations of the Japanese Kiban seismic network on land and 120 OBS stations of the S-net[27] (Fig. 1). We selected 2446 local earthquakes that occurred beneath the Kiban seismic network and 1311 events that were recorded by the S-net. From the high-quality vertical-component seismograms recorded by the S-net, we picked P-wave arrival times generated by the 1311 events (Supplementary Fig. 13). The picking accuracy of the arrival times is estimated to be 0.05–0.15 s. As a result, our data set contains a total of 109,890 P-wave arrival times from 3757 local earthquakes. Only arrival times of the events with well-located hypocenters (uncertainty < 3 km) are used in the tomographic inversion.

The tomographic method of Zhao et al.[47,48] is used to determine a detailed 3-D Vp model of the Tohoku subduction zone. The well-determined geometries of the Conrad and Moho discontinuities and the UBP are considered in the velocity

model[47,49]. Following Liu and Zhao[25], a 3-D grid with a lateral interval of 0.33° is set up to express the 3-D Vp structure in the study volume, and the vertical grid interval is 5 km around the UBP (Supplementary Fig. 5). Vp perturbations at the 3-D grid nodes from a starting Vp model are taken as unknown parameters. The Vp perturbation at any point in the study volume is calculated by linearly interpolating the Vp perturbations at the eight grid nodes surrounding that point. Considering the existence of strong velocity heterogeneities in the forearc area, we constructed a 3-D starting Vp model (Supplementary Fig. 14) by referring to results of active-source seismic surveys of the Tohoku forearc region[50]. Following previous tomographic studies in Tohoku, an initial 4% Vp perturbation is assigned to the subducting Pacific slab[23,24,47]. An efficient 3-D ray tracing technique[47] is used to calculate theoretical travel times and ray paths. Station elevations and the surface topography are taken into account in the 3-D ray tracing. A least-squares method with damping and smoothing regularizations[47] is used to solve the large but sparse system of observation equations that relate the arrival-time data to the unknown Vp parameters. The local earthquakes are relocated in the inversion process.

After the tomographic inversion, we calculated the 3-D Vp perturbations (Supplementary Fig. 3a) relative to an average 1-D velocity model (Supplementary Fig. 2). After the 3-D Vp model is obtained (Supplementary Figs. 3a and 15), we further determined a 2-D Vp model for the overriding Okhotsk plate beneath the Tohoku forearc (Supplementary Fig. 3d) using the method of Liu and Zhao[25]. For any point in the 2-D forearc area above the UBP, we calculate travel times $t_1$ and $t_3$ of a vertical ray path with a length d from the UBP to the Earth's surface for the 1-D Vp model (Supplementary Fig. 2) and the obtained 3-D Vp model, respectively. Then, the Vp anomaly at the point in the overriding plate is calculated using a simple relation $\frac{\delta V}{V_1} = \left( \frac{d}{t_3} - \frac{d}{t_1} \right) / \left( \frac{d}{t_1} \right)$. Supplementary Fig. 3d shows the obtained 2-D Vp image of the overriding Okhotsk plate.

To compare our Vp tomographic images with the results of residual topography and residual gravity[21], we determined residual Vp images from the obtained 3-D Vp model using a spectral averaging method that was designed to calculate the residual topography and residual gravity[21]. Arithmetic average of the Vp perturbations along each contour line of the UBP is calculated (Supplementary Fig. 3b, e), which is then subtracted from the original 3-D Vp model (Supplementary Fig. 3a, d). Thus, the residual Vp images along the UBP and in the overriding Okhotsk plate (ORP) are obtained (Supplementary Figs. 3c, f and 4).

By taking into account the complex velocity discontinuities (i.e., the Conrad, the Moho, and the subducting slab upper boundary), we can obtain more accurate ray paths and get a better tomographic model[23–25,47]. To evaluate whether uncertainties of these discontinuities could affect the main features of our tomographic model or not, we conducted a tomographic inversion with deeper velocity discontinuities. The inversion results (Supplementary Fig. 16) show very

similar features to those of Fig. 2 and Supplementary Fig. 3, indicating that the uncertainties of the velocity discontinuities have little effect on our tomographic result. The 1-D velocity model used in this study is determined based on the previous tomographic studies and active-source seismic surveys of NE Japan[23–25,47,48,50]. We have also calculated Vp perturbations using a different 1-D velocity model, and the results (Supplementary Fig. 17) suggest that the 1-D velocity model can change the Vp perturbations, but the residual velocity image shows the same pattern as that of Fig. 2. To investigate whether a low-V subducting oceanic crust affects our results or not, we conducted a tomographic inversion with a starting velocity model that includes a low-V oceanic crust atop the subducting Pacific slab. The results (Supplementary Fig. 18) show very similar features to those of Fig. 2 and Supplementary Fig. 3, indicating that the low-V oceanic crust has a very small effect on our results.

**Resolution tests**. Extensive checkerboard resolution tests (CRTs) are performed to evaluate the adequacy of ray coverage and the spatial resolution of our tomographic model (Supplementary Figs. 5–7). To perform a CRT, we first assigned positive and negative Vp perturbations to the adjacent grid nodes in the input model to calculate synthetic travel times with the same numbers of seismic stations, events, and ray paths as those in the real data set. To simulate picking errors of the arrival-time data, Gaussian noise (−0.2 to +0.2 s) with a standard deviation of 0.1 s is added to the synthetic travel times before the tomographic inversion. Supplementary Figs. 5–7 show the obtained CRT results with a lateral grid interval of 0.33°, which suggest that our 3-D Vp model has a spatial resolution of ~33 km in the lateral direction and ~5 km in depth in and around the UBP. We also performed several synthetic tests to evaluate the robustness of our tomographic model. The procedure of the synthetic tests is the same as that of the CRT except for the input model. The test results (Supplementary Figs. 8–10) indicate that the main features of our tomographic model are quite robust.

## Data availability
All data needed to evaluate the conclusions in the paper are present in the paper and/or the Supplementary Materials. Additional data related to this paper may be requested from the authors. The waveform data were downloaded freely from the Data Management Center of the Hi-net[28] and S-net[27] [http://www.hinet.bosai.go.jp/].

## Code availability
The free software GMT[51] and SAC[52] are used in this study. The analysis codes and related scripts for generating figures used in the main text and supplementary information are available from the corresponding authors upon reasonable request.

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

## Acknowledgements

We thank the data centers of the S-net, Hi-net, JMA Unified Earthquake Catalog, and Research Center for Prediction of Earthquakes and Volcanic Eruptions of Tohoku University for providing the high-quality waveform and arrival-time data used in this study. We are very grateful to Dr. Zhiteng Yu for his help for collecting the S-net arrival-time data. We appreciate helpful discussions with Drs. Xin Liu, Jianke Fan, Zewei Wang, Tao Gou, and Yaqian Liu. Prof. C. Satriano, H. Yue, and K. Satake kindly provided their coseismic slip models. This work was supported by research grants to D. Zhao from Japan Society for the Promotion of Science (No. 19H01996), Ministry of Education, Culture, Sports, Science and Techology (MEXT) of Japan under it's the Second Earthquake and Volcano Hazards Observation and Research Program, as well as the Core Research Cluster of Disaster Science in Tohoku University.

## Author contributions

D.Z. designed and led the project. Y.H. conducted data processing and inversion. Y.H. and D.Z. wrote the manuscript. G.T. and Y.X. contributed to the interpretations and preparation of the manuscript.

## Competing interests

The authors declare no competing interests.
