## [Peer Review File · Nature Communications]

Reviewers' comments:

Reviewer #1 (Remarks to the Author):

The 2011 Tohoku-oki earthquake is one of the largest earthquakes that are well recorded by modern instruments. Even though Japan has the best observations system, the source area of the earthquake rupture has not been covered by local seismometers when the earthquake occurred. The NIED set up kinds of OBS instrument under the Pacific Ocean, which provide unprecedented opportunity to determined high-resolution images of the mega thrust zone along the plate interface. The authors picked a lot of new arrival times from the oceanic events that were recorded by the OBS stations. The new data improves the resolution of seismic tomography under the Pacific Ocean very much. They found the great earthquake started at a point between down-dip high velocity anomaly and up-dip low velocity anomaly. New results provide essential information on seismotectonics of the subduction thrust zone.

I recommend a major revision for the present manuscript. My main concern of the seismic images is its robustness, given the variations of the starting velocity models. An advantage of the applied method the authors claim is that it solves complex velocity discontinuities. However, while the discontinuities have been well estimated under land areas, they are not well tested under the oceans. To what degree can the uncertainties in the velocity discontinuities (e.g., slab boundary, Moho in the forearm) affect the tomographic images.

A related issue is how the starting velocity model affects the tomographic images. How the used starting 1-D model determined. It is different with the JMA velocity model for earthquake location. Are the present seismic anomalies also notable when using JMA model? As for the subducting plate, many studies prove the existence of a oceanic crust at the top. The oceanic crust has very low velocities, but not in the starting model of this study. Will it change the patterns of velocity anomalies claimed in the present study?

The authors made routine checkerboard tests to show that the tomographic images are reliable. More synthetic tests of the high and low velocity patches close to the trench, in particular the stretch of the anomalies due to smearing effect, should be checked in details.

Line 190: What is the observational period of the OBS stations? As they were deployed after the Tohoku-oki earthquakes, the seismic images only reflect the structures after the great earthquake ruptures. Thus it is necessary to clarify whether the obtained images explain the mechanism of the earthquake formation. Are the features same as those before the earthquake rupture? Could the rupture, accompanying the heat during slips, change the seismic velocities along the fault plane? Have you tried to revealed the seismic images in different periods so that to check possible temporal change of seismic anomalies at the great thrust zone?

Figure 1: Why are there few earthquakes located near the rupture area of the Tohoku-oki earthquake? Is it due to actually weak seismicity? Or the waveforms are too noisy for phase picking?
Figure S7: Have you filtered the data?

Reviewer #2 (Remarks to the Author):

Constraining volumetric structural heterogeneity around the rupture zone of great megathrust ruptures is essential for addressing the degree to which physical properties of the surrounding rock/sediments influence the rupture dynamics, aftershocks, and post-seismic processes. Often, 3D tomography of the offshore megathrust environment is severely limited in path coverage. Fortunately, the S-net now provides very good coverage extending to near the Japan trench, enabling the highest quality shallow tomography yet performed for any great rupture source zone. This builds upon prior fairly well resolved imaging of the deeper half of the rupture zone, with similar basic findings for that region. The shallow region of very large slip extending to the trench is found to have low P-wave velocity along and above the megathrust boundary. This suggests that mechanical properties (low rigidity, plausibly high porosity) are present and likely play an important role in the failure and up-dip rupture properties for the 2011 event. This is compatible with sparser measurements of the shallow portion of other megathrust faults, indicating that such material properties are important for large thrust events in general. This paper relates the spatial distribution of relatively low or relatively high velocity material near the fault to slip magnitude, generation of short-period radiation during slip, and relocking/viscoelastic deformation versus afterslip.

I think the tomography is impressive and exploits the unique opportunity provided by the S-net recordings of many regional earthquakes, supplemented by on-land observations. Thus, the tomography model is better resolved in the offshore area than any prior work in this region. Given the spatial correspondence of low P velocity, large slip, and lack of coherent bursts of short-period seismic radiation from the megathrust up-dip of the hypocenter, the inferences about mechanical controls on the dynamic failure process are credible and, for this reviewer, likely. The paper does not include any self-consistent dynamic rupture modeling to validate some of the claims of having made progress on the 'causal' influences of the mechanical regime, but I expect it will hold up to modeling, although there is still going to be a lot of uncertainty in dynamic weakening processes. Thus, I think the paper is quite valuable, but it does tend to overstate the degree to which we have improved quantitative understanding of the rupture dynamics (correlation is not causation).

The paper was rather uneven in writing; I felt compelled to make many editorial suggestions in the main text, and include a tracked-changes pdf version to try to improve the writing. There was notable and undesirable redundancy in some statements, and I tried to eliminate some of that. I also suggest a few references to include; one is a large review paper I wrote discussing all of the published models and limitations of various data used in various models; this paper shows only one slip model, which is a reasonable model, but there are many solutions that are at least equally good that have somewhat different spatial pattern given that such inversions are very non-unique and depend on the method and data used. The other reference I suggest is a discussion of the region down-dip along-strike to the north of the 2011 coseismic slip zone where variations in prior earthquake occurrence and afterslip show clear correspondence to the low velocity region in the

deepertomographic image, especially along the Sanriku coastline. I am not trolling for citations, but these references do seem relevant. I also edited a lot of English grammar. I hope the authors will not take any offense at my editorial suggestions, which are intended to improve clarity and to simplify some of the statements.

I do not have any issues with the technical aspects of the tomography; this appears to be well done and the authors have excellent competency. Overall, the paper, after suitable editing, is appropriate for publication in Nature Communications due to the important contribution to imaging the velocity heterogeneity around the up-dip 2011 slip zone. My final suggestion is to reemphasize that constraining volumetric material properties is an important step in quantifying rupture dynamics, but it does not constitute a physical model of the dynamic properties, so the authors might be a bit more cautious about claiming that causal processes have actually been revealed by the tomography.

Thorne Lay

Review of NCOMMS-19-27127-T – Tomography of the source zone of the great 2011 Tohoku earthquake

This manuscript integrates passive seismic data recorded onshore (Hi-net) and offshore (S-net) to perform seismic travel-time tomography for the NE Japan subduction. Particular emphasis is given to resolving structure around the 2011 Mw 9.0 Tohoku-oki earthquake, and trying to resolve the physical controls on the distribution of large co-seismic displacements and high-frequency radiation. The primary conclusion is that the 2011 earthquake nucleated from a boundary between high and low velocities, and that low velocities reflect low rigidity materials that were unable to arrest rupture propagation, or produce high frequency radiation.

This earthquake and subduction zone are of wide interest, so in a topical sense, I think this study would be of interest to the readers of Nature Communications. Having said that, there are several issues that I feel should be considered by the authors before this manuscript is accepted for publication. My primary suggestions for improvement are: 1) Show cross-sections through the model; 2) Show the model as absolute wavespeeds, rather than velocity anomalies; 3) Expand discussion of the physical interpretation of wavespeeds.

Below I have included a bit more detail on each of the points noted above.

Main comments:

- 1) It would be good to see some cross-sections through the model so readers can evaluate how much of the variation in wavespeeds along the megathrust are due to vertical smearing from the overthrusting and subducting plates. In the region of primary interest, the Tohoku earthquake zone, the slices through the seismic velocity model taken above (ORP) and along (UBP) the subduction interface are really quite similar. This would indicate to me, that what the authors are actually resolving is variations in the structure of the hanging wall. I am also sceptical about the ability of the inversion, with 5 km vertical node increments, to independently resolve structure along the fault zone, in which case showing UBP and ORP slices is unnecessary. In any case, showing Vp cross-sections at various locations along strike is, in my view, a necessary and valuable addition.
- 2) I would recommend showing the Vp model as absolute wavespeeds, rather than wavespeeds anomalies. Is these possible to reconstruct by applying the wavespeed anomalies derived from your inversion back into the starting model? My reason for requesting this, is that the starting model in figure S8 shows to significant margin-normal variations in wavespeeds in the overthrusting plate. This makes it hard to compare wavespeed variations down-dip, because they are normalised against different starting values. Reverting back to absolute wavespeeds would greatly simplify the physical interpretation of the tomographic model.
- 3) My final request is that the authors be slightly more assertive and direct about some of the physical interpretations that they use to link their inversion results with the source characteristics of the Tohoku-oki earthquake. For example, what does the high-to-low Vp transition physically represent? Is this a backstop separating crustal rocks from accreted sediments, or is this an inherited transition in upper plate lithology? The high Vp anomaly is

described as an asperity, but this is a frictional characteristic, so there needs to be some physical interpretation to link Vp structure and fault zone friction. The low-Vp area near the trench is described as having low rigidity, but would this not cause the outer-wedge to deform internally, rather than slip co-seismically, when pushed by failure of the deeper seismogenic zone? I am not suggesting the authors interpretation is incorrect, but I think the text could be improved to be more direct about their interpretations.

Minor comments and typographical suggestions:

Line 30-31 – don't dynamic rupture models show rupture fronts arresting when it enters weak, compliant or unloaded materials (e.g. Tinti, Bizzarri and Cocco, Ann. Geophysics, 2005)

Line 82 – how much of the wavespeed variation in this slice is generated in the overthrusting plate?

Line 114 – even if clay materials cannot accumulate elastic stress, does this preclude them from slipping if pushed to failure by large slip along the deeper seismogenic zone?

Line 124 – what is the physical cause of the frictional asperity?

Line 127 – replace "rigid" with "rigidity"

Line 171-172 – Does your model provide any insight into what is driving this variation? Subduction inputs, or hanging-wall lithology?

Responses to the review comments

(The black words show the review comments; **the blue words show our responses**)

Reviewer #1 comments:

The 2011 Tohoku-oki earthquake is one of the largest earthquakes that are well recorded by modern instruments. Even though Japan has the best observations system, the source area of the earthquake rupture has not been covered by local seismometers when the earthquake occurred. The NIED set up kinds of OBS instrument under the Pacific Ocean, which provide unprecedented opportunity to determined high-resolution images of the megathrust zone along the plate interface. The authors picked a lot of new arrival times from the oceanic events that were recorded by the OBS stations. The new data improves the resolution of seismic tomography under the Pacific Ocean very much. They found the great earthquake started at a point between down-dip high velocity anomaly and up-dip low velocity anomaly. New results provide essential information on seismotectonics of the subduction thrust zone.

Thank you very much for your positive comments!

1. I recommend a major revision for the present manuscript. My main concern of the seismic images is its robustness, given the variations of the starting velocity models. An advantage of the applied method the authors claim is that it solves complex velocity discontinuities. However, while the discontinuities have been well estimated under land areas, they are not well tested under the oceans. To what degree can the uncertainties in the velocity discontinuities (e.g., slab boundary, Moho in the forearm) affect the tomographic images.

Nice point! By taking into account the velocity discontinuities, we can obtain more accurate ray paths and get a better tomographic model. The velocity discontinuities we used in this study are constructed based on several previous seismic studies, which are quite reliable (with depth uncertainties < 3 km). They have been used by many previous tomographic studies (e.g., *Zhao et al.*, 2011; *Huang & Zhao*, 2013; *Liu & Zhao*, 2018). **We have also conducted a tomographic inversion with velocity discontinuities that are 3 km deeper than their original depths.** The inversion results (Figure S16) show a similar pattern to those shown in Fig. 2 and Fig. S3, suggesting that the uncertainties of velocity discontinuities do not affect the main features of our tomographic model. We have made this point clear in our revised manuscript. Please see lines 268-275.

2. A related issue is how the starting velocity model affects the tomographic images. How the used starting 1-D model determined. It is different with the JMA velocity model for earthquake location. Are the present seismic anomalies also notable when using JMA model? As for the subducting plate, many studies prove the existence of an oceanic crust at the top. The oceanic crust has very low velocities, but not in the starting model of this study. Will it change the patterns of velocity anomalies claimed in the present study?

To make the earthquake relocation more accurate, we have adopted a 3-D starting velocity model (Fig. S14). After the tomographic inversion, we calculated the velocity perturbations at the 3-D grid nodes relative to a simple 1-D velocity model (Fig. S2) and got the tomographic results (Fig. S3). The 1-D velocity model used in this study is determined based on previous tomographic studies and active-source seismic surveys of the Tohoku arc. The effect of the subducting Pacific slab is not fully taken into account in the 1-D JMA velocity model. **To evaluate whether the 1-D velocity model affects the tomographic pattern or not, we have also calculated velocity perturbations at the 3-D grid nodes using a different 1-D velocity model**, in which V_p is 5.5 km/s in the upper crust and 6.2 km/s in the lower crust. The results (Fig. S17) show that the main features are similar to those of our preferred tomographic model (Fig. S3), suggesting that the reference 1-D velocity model does not affect our tomographic result significantly, especially the residual V_p image (Fig. 2).

To investigate whether a low-velocity (low- V) oceanic crust atop the subducting Pacific slab affects our results or not, we conducted a tomographic inversion with a starting velocity model that includes a low- V oceanic crust atop the slab. The inversion results (Fig. S18) show that main features of the residual V_p image are the same as those in Fig. 2.

In summary, although the 1-D V_p model and a low- V oceanic crust in the starting model slightly affect the absolute V_p images, the pattern of our residual V_p image (Fig. 2) remains the same. We have made these points clear in the revised manuscript, please see lines 275-285.

3. The authors made routine checkerboard tests to show that the tomographic images are reliable. More synthetic tests of the high and low velocity patches close to the trench, in particular the stretch of the anomalies due to smearing effect, should be checked in details.

Nice point! **We have newly conducted several synthetic tests to evaluate the robustness of our tomographic model.** The results of these synthetic tests (Fig. S8-10) indicate that the main

features of our tomographic model are quite reliable. Please see Fig. S8-10 and lines 298-301 in our revised manuscript.

4. Line 190: What is the observational period of the OBS stations? As they were deployed after the Tohoku-oki earthquakes, the seismic images only reflect the structures after the great earthquake ruptures. Thus, it is necessary to clarify whether the obtained images explain the mechanism of the earthquake formation. Are the features same as those before the earthquake rupture? Could the rupture, accompanying the heat during slips, change the seismic velocities along the fault plane? Have you tried to revealed the seismic images in different periods so that to check possible temporal change of seismic anomalies at the great thrust zone?

The S-net stations were deployed several years after the 2011 Tohoku-oki earthquake. Seismograms recorded at the S-net stations have been available since August 2016 to now. Hence, our tomographic images reflect seismic velocity heterogeneities after the great 2011 Tohoku-oki earthquake. The frictional heat released during the March 2011 coseismic slips could affect seismic velocity along the megathrust zone. However, *Fulton et al.* (2013) showed that the amount of frictional heat generated by the Tohoku rupture was extremely small. In addition, seismic velocity of rocks and minerals is mainly controlled by the lithology of materials at shallow depths (*Hacker et al.*, 2003). Thus, we think that the 2011 Tohoku-oki earthquake rupture could not cause a great effect on the velocity structure of the megathrust zone. The very small (meter-scale) change of velocity structure caused by the rupture, if any, could not be imaged due to the limited spatial resolution (kilometer-scale) of our tomographic model. We have made this point clear in our revised manuscript, please see lines 112-124.

Because seismograms recorded at the S-net stations are available from August 2016, we think that our travel-time data are insufficient to reveal seismic images in different periods. This is a nice suggestion, and we will try it when we have enough travel-time data in the near future.

5. Figure 1: Why are there few earthquakes located near the rupture area of the Tohoku-oki earthquake? Is it due to actually weak seismicity? Or the waveforms are too noisy for phase picking?

There is actually weak seismicity there. After the 2011 Tohoku earthquake, large shear stress reduction occurred in the mainshock rupture zone, and it will need time for reaccumulation of sufficient stress to rupture that area (*Wetzler et al.*, 2018).

6. Figure S7: Have you filtered the data?

Yes, we have filtered the waveforms using a Butterworth filter in a frequency band of 1-10 Hz. We have made it clear in our revised manuscript. Please see the Fig. S13 caption.

Reviewer #2 comments:

Constraining volumetric structural heterogeneity around the rupture zone of great megathrust ruptures is essential for addressing the degree to which physical properties of the surrounding rock/sediments influence the rupture dynamics, aftershocks, and post-seismic processes. Often, 3D tomography of the offshore megathrust environment is severely limited in path coverage. Fortunately, the S-net now provides very good coverage extending to near the Japan trench, enabling the highest quality shallow tomography yet performed for any great rupture source zone. This builds upon prior fairly well resolved imaging of the deeper half of the rupture zone, with similar basic findings for that region. The shallow region of very large slip extending to the trench is found to have low P-wave velocity along and above the megathrust boundary. This suggests that mechanical properties (low rigidity, plausibly high porosity) are present and likely play an important role in the failure and up-dip rupture properties for the 2011 event. This is compatible with sparser measurements of the shallow portion of other megathrust faults, indicating that such material properties are important for large thrust events in general. This paper relates the spatial distribution of relatively low or relatively high velocity material near the fault to slip magnitude, generation of short-period radiation during slip, and relocking/viscoelastic deformation versus afterslip.

Dear Prof. Lay: Thank you very much!

I think the tomography is impressive and exploits the unique opportunity provided by the S-net recordings of many regional earthquakes, supplemented by on-land observations. Thus, the tomography model is better resolved in the offshore area than any prior work in this region. Given the spatial correspondence of low P velocity, large slip, and lack of coherent bursts of short-period seismic radiation from the megathrust up-dip of the hypocenter, the inferences about mechanical controls on the dynamic failure process are credible and, for this reviewer, likely. The paper does not include any self-consistent dynamic rupture modeling to validate some of the claims of having made progress on the 'causal' influences of the mechanical regime, but I expect it will hold up to modeling, although there is still going to be a lot of uncertainty in dynamic weakening processes. Thus, I think the paper is

quite valuable, but it does tend to overstate the degree to which we have improved quantitative understanding of the rupture dynamics (correlation is not causation).

Nice point! We have revised our manuscript and toned down our discussions about the rupture dynamics, in particular, we have revised the descriptions about the “causal mechanism”. Please see lines 24, 202-204, and 213-216 in our revised manuscript.

The paper was rather uneven in writing; I felt compelled to make many editorial suggestions in the main text, and include a tracked-changes pdf version to try to improve the writing. There was notable and undesirable redundancy in some statements, and I tried to eliminate some of that. I also suggest a few references to include; one is a large review paper I wrote discussing all of the published models and limitations of various data used in various models; this paper shows only one slip model, which is a reasonable model, but there are many solutions that are at least equally good that have somewhat different spatial pattern given that such inversions are very non-unique and depend on the method and data used. The other reference I suggest is a discussion of the region down-dip along-strike to the north of the 2011 coseismic slip zone where variations in prior earthquake occurrence and afterslip show clear correspondence to the low velocity region in the deeper tomographic image, especially along the Sanriku coastline. I am not trolling for citations, but these references do seem relevant. I also edited a lot of English grammar. I hope the authors will not take any offense at my editorial suggestions, which are intended to improve clarity and to simplify some of the statements.

Thank you very much for kindly improving our English expressions. We have added the references that you suggested to our revised manuscript. Other slip models have also been included to the revised manuscript. Please see lines 142-144 and Fig. S12. We have also added a discussion about the region down-dip along-strike to the north of the 2011 coseismic slip zone. See lines 102-105.

I do not have any issues with the technical aspects of the tomography; this appears to be well done and the authors have excellent competency. Overall, the paper, after suitable editing, is appropriate for publication in Nature Communications due to the important contribution to imaging the velocity heterogeneity around the up-dip 2011 slip zone. My final suggestion is to reemphasize that constraining volumetric material properties is an important step in quantifying rupture dynamics, but it does not constitute a physical model of the dynamic properties, so the authors might be a bit more cautious about claiming that causal processes have actually been revealed by the tomography.

Nice point! We have toned down our discussions about the rupture dynamics and mechanism in the revised manuscript.

Thorne Lay

Thank you very much again!

Reviewer #3 comments:

This manuscript integrates passive seismic data recorded onshore (Hi-net) and offshore (S-net) to perform seismic travel-time tomography for the NE Japan subduction. Particular emphasis is given to resolving structure around the 2011 Mw 9.0 Tohoku-oki earthquake, and trying to resolve the physical controls on the distribution of large co-seismic displacements and high-frequency radiation.

The primary conclusion is that the 2011 earthquake nucleated from a boundary between high and low velocities, and that low velocities reflect low rigidity materials that were unable to arrest rupture propagation, or produce high frequency radiation. This earthquake and subduction zone are of wide interest, so in a topical sense, I think this study would be of interest to the readers of Nature Communications. Having said that, there are several issues that I feel should be considered by the authors before this manuscript is accepted for publication. My primary suggestions for improvement are: 1) Show cross-sections through the model; 2) Show the model as absolute wave speeds, rather than velocity anomalies; 3) Expand discussion of the physical interpretation of wave speeds. Below I have included a bit more detail on each of the points noted above.

Thank you very much! We have very carefully revised our manuscript following all of your suggestions. Please see the followings for details.

Main comments:

1. It would be good to see some cross-sections through the model so readers can evaluate how much of the variation in wave speeds along the megathrust are due to vertical smearing from the overthrusting and subducting plates. In the region of primary interest, the Tohoku earthquake zone, the slices through the seismic velocity model taken above (ORP) and along (UBP) the subduction interface are really quite similar. This would indicate to me, that what the authors are actually resolving is variations in the structure of the hanging wall. I am also sceptical about the ability of the inversion, with 5 km vertical node increments, to independently resolve structure along the fault zone, in which case showing UBP and ORP slices is unnecessary. In any case, showing V_p

cross-sections at various locations along strike is, in my view, a necessary and valuable addition.

Nice point! We have added three vertical cross-sections of our results, please see Fig. S15 in the revised manuscript.

To confirm the robustness of our tomographic model, we have newly conducted several synthetic tests. The test results (Fig. S8-S10) indicate that our tomographic model is quite reliable. We acknowledge that the tomographic image along the UBP may reflect not only structural heterogeneities along the megathrust zone but also lithologic changes in the upper plate. Following your suggestion, we have moved the ORP image to the supporting information (Fig. S1). See lines 86-87.

2. I would recommend showing the V_p model as absolute wave speeds, rather than wave speeds anomalies. Is these possible to reconstruct by applying the wave speed anomalies derived from your inversion back into the starting model? *My reason for requesting this, is that the starting model in figure S8 shows to significant margin-normal variations in wave speeds in the overthrusting plate.* This makes it hard to compare wave speed variations down-dip, because they are normalised against different starting values. Reverting back to absolute wave speeds would greatly simplify the physical interpretation of the tomographic model.

To make earthquake relocation more accurate, we adopt a 3-D starting velocity model (Fig. S14). After the tomographic inversion, we calculated the V_p perturbations relative to a reference 1-D velocity model (Fig. S2), and the tomographic images are obtained after the arithmetic average is removed (Fig. S3). **Thus, the V_p perturbations are actually relative to the 1-D velocity model (Fig. S2).** We have made it clear in the revised manuscript. See lines 249-251.

We have also added three vertical cross-sections showing the absolute 3-D V_p model (Fig. S15) in the revised manuscript.

3. My final request is that the authors be slightly more assertive and direct about some of the physical interpretations that they use to link their inversion results with the source characteristics of the Tohoku-oki earthquake.

For example, what does the high-to-low V_p transition physically represent? Is this a backstop separating crustal rocks from accreted sediments, or is this an inherited transition in upper plate lithology?

Nice Point! *Nakamura et al. (2014)* imaged the backstop interface near the Japan trench, which

is shallower than the location of high-to-low Vp transition. Hence, the along-dip high-to-low Vp transition is not a backstop interface. We think that the along-dip high-to-low Vp transition may reflect not only fault friction transition along the megathrust zone but also lithologic change in the upper plate. We have made this point clear in the revised manuscript. Please see lines 182-188.

The high Vp anomaly is described as an asperity, but this is a frictional characteristic, so there needs to be some physical interpretation to link Vp structure and fault zone friction.

Typical crystalline rocks comprising the volcanic arc are characterized by a relatively high static coefficient of friction and a high Vp, whereas increasing clay content can reduce the coefficient of friction and seismic velocity (*Bassett et al., 2016; Morrow et al., 2000*). The high-V anomaly corresponds to the high residual gravity anomaly, which is interpreted as granite batholiths (*Bassett et al., 2016*). The frictional asperity may be caused by the granite batholiths. The high-to-low Vp transition may reflect not only the lithologic change in the upper plate but also the fault friction transition along the megathrust zone. We have described it in the revised manuscript. Please see lines 147-148 and 182-188.

The low-Vp area near the trench is described as having low rigidity, but would this not cause the outer-wedge to deform internally, rather than slip co-seismically, when pushed by failure of the deeper seismogenic zone?

Nice point! Using seismic reflection data, *Koarira et al. (2012)* revealed a deformed upheaval structure at shallow depth. Considering the existence of low-friction clay along the megathrust zone (*Chester et al., 2013; Fulton et al., 2013; Ujiie et al., 2013*), we think that the low-Vp anomaly with low rigidity at shallow depth may cause not only large coseismic slip near the trench but also the outer-wedge deformation. We have added this point to the revised manuscript. See lines 153-154.

I am not suggesting the authors interpretation is incorrect, but I think the text could be improved to be more direct about their interpretations.

Thank you very much. We have improved the related parts.

Minor comments and typographical suggestions:

4. Line 30-31 – don't dynamic rupture models show rupture fronts arresting when it enters weak, compliant or unloaded materials (e.g. Tinti, Bizzarri and Cocco, *Ann. Geophysics*, 2005)

Yes, you are right. We have changed the sentence to “*The slow anomaly at shallow depths near the Japan trench may reflect low-rigidity materials that are close to the free surface, resulting in large coseismic slips and weak high-frequency radiation*”. See lines 30-32 in the revised manuscript.

5. Line 82 – how much of the wave speed variation in this slice is generated in the overthrusting plate?

We have newly performed several synthetic tests (Fig. S8-10), which indicate that the smearing effect around the UBP is not significant.

We estimate that over ~75% of velocity perturbations along the UBP result from the top portion of the subducting Pacific slab, and less than ~25% of the velocity perturbations result from the overriding plate.

6. Line 114 – even if clay materials cannot accumulate elastic stress, does this preclude them from slipping if pushed to failure by large slip along the deeper seismogenic zone?

We have slightly modified this sentence, please see lines 136-138.

7. Line 124 – what is the physical cause of the frictional asperity?

The high-V anomaly corresponds to the high residual gravity anomaly, which is interpreted as granite batholiths (Bassett et al., 2016). Hence, the frictional asperity may be produced by the granite batholiths. We have mentioned it at lines 147-148 of the revised manuscript.

8. Line 127 – replace “rigid” with “rigidity”

Done.

9. Line 171-172 – Does your model provide any insight into what is driving this variation? Subduction inputs, or hanging-wall lithology?

This variation may be caused by not only changes of friction and material properties along the Tohoku megathrust zone but also the lithologic transition in the overriding plate. We have made this point clear, please see lines 199-201 in the revised manuscript.

Thank you very much again!

Related references

- Bassett, D., Sandwell, D. T., Fialko, Y., & Watts, A. B. (2016). Upper-plate controls on co-seismic slip in the 2011 magnitude 9.0 Tohoku-oki earthquake. *Nature*, *531*(7592), 92–96. <https://doi.org/10.1038/nature16945>
- Chester, F. M., Rowe, C., Ujiie, K., Kirkpatrick, J., Regalla, C., Remitti, F., et al. (2013). Structure and composition of the plate-boundary slip zone for the 2011 Tohoku-oki earthquake. *Science*, *342*(6163), 1208–1211. <https://doi.org/10.1126/science.1243719>
- Fulton, P. M., Brodsky, E. E., Kano, Y., Mori, J., Chester, F., Ishikawa, T., et al. (2013). Low coseismic friction on the Tohoku-oki fault determined from temperature measurements. *Science*, *342*(6163), 1214–1217. <https://doi.org/10.1126/science.1243641>
- Hacker, B. R., Abers, G. A., & Peacock, S. M. (2003). Subduction factory 1. Theoretical mineralogy, densities, seismic wave speeds, and H₂O contents. *Journal of Geophysical Research: Solid Earth*, *108*(B1). <https://doi.org/10.1029/2001JB001127>
- Huang, Z., & Zhao, D. (2013). Mechanism of the 2011 Tohoku-oki earthquake (Mw 9.0) and tsunami: Insight from seismic tomography. *Journal of Asian Earth Sciences*, *70–71*, 160–168. <https://doi.org/10.1016/j.jseae.2013.03.010>
- Huang, Z., Zhao, D., & Wang, L. (2011). Seismic heterogeneity and anisotropy of the Honshu arc from the Japan Trench to the Japan Sea: Heterogeneity and anisotropy of Honshu arc. *Geophysical Journal International*, *184*(3), 1428–1444. <https://doi.org/10.1111/j.1365-246X.2011.04934.x>
- Liu, X., & Zhao, D. (2018). Upper and lower plate controls on the great 2011 Tohoku-oki earthquake. *Science Advances*, *4*(6), eaat4396. <https://doi.org/10.1126/sciadv.aat4396>
- Morrow, C. A., Moore, D. E., & Lockner, D. A. (2000). The effect of mineral bond strength and adsorbed water on fault gouge frictional strength. *Geophysical Research Letters*, *27*(6), 815–818. <https://doi.org/10.1029/1999GL008401>
- Nakamura, Y., Kodaira, S., Cook, B. J., Jeppson, T., Kasaya, T., Yamamoto, Y., et al. (2014). Seismic imaging and velocity structure around the JFAST drill site in the Japan Trench: low V_p, high V_p/V_s in the transparent frontal prism. *Earth, Planets and Space*, *66*(1), 121. <https://doi.org/10.1186/1880-5981-66-121>
- Ujiie, K., Tanaka, H., Saito, T., Tsutsumi, A., Mori, J. J., Kameda, J., et al. (2013). Low coseismic shear stress on the Tohoku-oki megathrust determined from laboratory experiments. *Science*, *342*(6163), 1211–1214. <https://doi.org/10.1126/science.1243485>

- Wetzler, N., Lay, T., Brodsky, E. E., & Kanamori, H. (2018). Systematic deficiency of aftershocks in areas of high coseismic slip for large subduction zone earthquakes. *Science Advances*, 4(2), eaao3225. <https://doi.org/10.1126/sciadv.aao3225>
- Zhao, D., Hasegawa, A., & Horiuchi, S. (1992). Tomographic imaging of P and S wave velocity structure beneath northeastern Japan. *Journal of Geophysical Research*, 97(B13), 19909–19928. <https://doi.org/10.1029/92JB00603>
- Zhao, D., Hasegawa, A., & Kanamori, H. (1994). Deep structure of Japan subduction zone as derived from local, regional, and teleseismic events. *Journal of Geophysical Research*, 99(B11), 22313–22329. <https://doi.org/10.1029/94JB01149>
- Zhao, D., Huang, Z., Umino, N., Hasegawa, A., & Kanamori, H. (2011). Structural heterogeneity in the megathrust zone and mechanism of the 2011 Tohoku-oki earthquake (Mw 9.0). *Geophysical Research Letters*, 38(17). <https://doi.org/10.1029/2011GL048408>

REVIEWERS' COMMENTS:

Reviewer #1 (Remarks to the Author):

The authors have conducted more inversions and confirmed that the major features are robust. I think the present revision is all right for publication.

Reviewer #2 (Remarks to the Author):

The revised manuscript has addressed the key issues raised by the various reviewers, including all of my comments and suggestions in the annotated manuscript. The issue of whether the tomography images a perturbed state after the great earthquake rupture is raised and partially addressed. I agree that there was not a huge thermal perturbation from sliding friction, but that is not what I would expect to be the primary perturbation. I would rather expect damaged; induced fracturing of the region surrounding the 60 m slip zone, to possibly impact velocity heterogeneity. I do believe that the lag time prior to deployment of the S-net allows much of this effect to heal, although various time scales may be involved. Anyhow, this is just an additional nuance to the response to this question that the authors may wish to make. I find the paper suitable for publication as is.

Thorne Lay

Reviewer #3 (Remarks to the Author):

First off, please let me apologise for the time it has taken me to complete this review. Unfortunately this review coincided with an extremely busy period for my family, but I did not want to pass up the opportunity to reevaluate this important manuscript.

I have reviewed the revised manuscript, supplementary information and the authors response to the comments of all reviewers. Overall, I think the authors should be commended for the professional manner in which they considered and incorporated all reviewers comments. The improvements to the manuscript are significant, and have greatly improved the readability and potential impact of this work.

I have no further suggested changes, and feel this manuscript is acceptable for publication.

Dan Bassett

Responses to the review comments

(The black words show the review comments; **the blue words show our responses**)

Reviewer #1 comments:

The authors have conducted more inversions and confirmed that the major features are robust. I think the present revision is all right for publication.

Thank you very much for your positive comments!

Reviewer #2 comments:

The revised manuscript has addressed the key issues raised by the various reviewers, including all of my comments and suggestions in the annotated manuscript. The issue of whether the tomography images a perturbed state after the great earthquake rupture is raised and partially addressed. I agree that there was not a huge thermal perturbation from sliding friction, but that is not what I would expect to be the primary perturbation. I would rather expect damaged; induced fracturing of the region surrounding the 60 m slip zone, to possibly impact velocity heterogeneity. I do believe that the lag time prior to deployment of the S-net allows much of this effect to heal, although various time scales may be involved. Anyhow, this is just an additional nuance to the response to this question that the authors may wish to make. I find the paper suitable for publication as is.

Thorne Lay

Dear Prof. Lay: Thank you very much!

Reviewer #3 comments:

First off, please let me apologise for the time it has taken me to complete this review. Unfortunately this review coincided with an extremely busy period for my family, but I did not want to pass up the opportunity to reevaluate this important manuscript.

I have reviewed the revised manuscript, supplementary information and the authors response to the comments of all reviewers. Overall, I think the authors should be commended for the professional manner in which they considered and incorporated all reviewers comments. The improvements to the manuscript are significant, and have greatly improved the readability and potential impact of this work.

I have no further suggested changes, and feel this manuscript is acceptable for publication.

Dan Bassett

Dear Prof. Bassett: Thank you very much!